# signSGD with Majority Vote is Communication Efficient And Fault Tolerant

**Jeremy Bernstein**[1]*, **Jiawei Zhao**[12]*, **Kamyar Azizzadenesheli**[3], **Anima Anandkumar**[1]
[1]Caltech, [2]Nanjing University of Aeronautics and Astronautics, [3]UC Irvine
bernstein@caltech.edu, jiaweizhao@nuaa.edu.cn,
kazizzad@uci.edu, anima@caltech.edu

## Abstract

Training neural networks on large datasets can be accelerated by distributing the workload over a network of machines. As datasets grow ever larger, networks of hundreds or thousands of machines become economically viable. The time cost of communicating gradients limits the effectiveness of using such large machine counts, as may the increased chance of network faults. We explore a particularly simple algorithm for robust, communication-efficient learning—signSGD. Workers transmit only the sign of their gradient vector to a server, and the overall update is decided by a majority vote. This algorithm uses $32\times$ less communication per iteration than full-precision, distributed SGD. Under natural conditions verified by experiment, we prove that signSGD converges in the large *and* mini-batch settings, establishing convergence for a parameter regime of Adam as a byproduct. Aggregating sign gradients by majority vote means that no individual worker has too much power. We prove that unlike SGD, majority vote is robust when up to 50% of workers behave adversarially. The class of adversaries we consider includes as special cases those that invert or randomise their gradient estimate. On the practical side, we built our distributed training system in Pytorch. Benchmarking against the state of the art collective communications library (NCCL), our framework—with the parameter server housed entirely on one machine—led to a 25% reduction in time for training `resnet50` on Imagenet when using 15 AWS `p3.2xlarge` machines.

## 1 Introduction

The most powerful supercomputer in the world is currently a cluster of over 27,000 GPUs at Oak Ridge National Labs (TOP500, 2018). Distributed algorithms designed for such large-scale systems typically involve both computation and communication: worker nodes compute intermediate results locally, before sharing them with their peers. When devising new machine learning algorithms for distribution over networks of thousands of workers, we posit the following desiderata:

**D1** fast algorithmic convergence;  **D3** communication efficiency;

**D2** good generalisation performance;  **D4** robustness to network faults.

When seeking an algorithm that satisfies all four desiderata **D1–4**, inevitably some tradeoff must be made. Stochastic gradient descent (SGD) naturally satisfies **D1–2**, and this has buoyed recent advances in deep learning. Yet when it comes to large neural network models with hundreds of millions of parameters, distributed SGD can suffer large communication overheads. To make matters worse, any faulty SGD worker can corrupt the entire model at any time by sending an infinite gradient, meaning that SGD without modification is not robust.

A simple algorithm with aspirations towards all desiderata **D1–4** is as follows: workers send the sign of their gradient up to the parameter server, which aggregates the signs and sends back only the majority decision. We refer to this algorithm as signSGD with majority vote. All communication

---

*JB was primary contributor for theory. JZ was primary contributor for large-scale experiments.

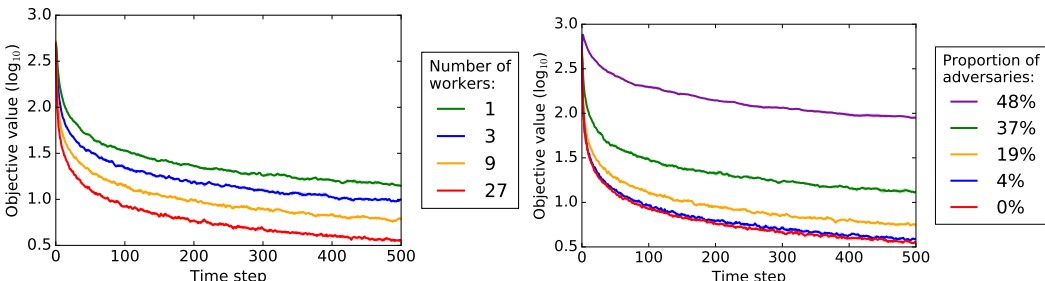

Figure 1: Toy experiments. SIGNSGD with majority vote is run on a 1000-dimensional quadratic with $\mathcal{N}(0, 1)$ noise added to each gradient component. Adversarial experiments are run with 27 total workers. These plots may be reproduced in a web browser by running this Jupyter notebook.

to *and* from the parameter server is compressed to one bit, so the algorithm certainly gives us **D3**. What's more, in deep learning folklore sign based methods are known to perform well, indeed inspiring the popular RMSPROP and ADAM optimisers (Balles & Hennig, 2018), giving hope for **D1**. As far as robustness goes, aggregating gradients by a majority vote denies any individual worker too much power, suggesting it may be a natural way to achieve **D4**.

In this work, we make the above aspirations rigorous. Whilst **D3** is immmediate, we provide the first convergence guarantees for SIGNSGD in the mini-batch setting, providing theoretical grounds for **D1**. We show how theoretically the behaviour of SIGNSGD changes as gradients move from high to low signal-to-noise ratio. We also extend the theory of majority vote to show that it achieves a notion of Byzantine fault tolerance. A distributed algorithm is *Byzantine fault tolerant* (Blanchard et al., 2017) if its convergence is robust when up to 50% of workers behave adversarially. The class of adversaries we consider contains interesting special cases, such as robustness to a corrupted worker sending random bits, or a worker that inverts their gradient estimate. Though our adversarial model is not the most general, it is interesting as a model of network faults, and so gives us **D4**.

Next, we embark on a large-scale empirical validation of our theory. We implement majority vote in the Pytorch deep learning framework, using CUDA kernels to bit pack sign tensors down to one bit. Our results provide experimental evidence for **D1–D4**. Comparing our framework to NCCL (the state of the art communications library), we were able to speed up Imagenet training by 25% when distributing over 7 to 15 AWS `p3.2xlarge` machines, albeit at a slight loss in generalisation.

Finally, in an interesting twist, the theoretical tools we develop may be brought to bear on a seemingly unrelated problem in the machine learning literature. Reddi et al. (2018) proved that the extremely popular ADAM optimiser in general does not converge in the mini-batch setting. This result belies the success of the algorithm in a wide variety of practical applications. SIGNSGD is equivalent to a special case of ADAM, and we establish the convergence rate of mini-batch SIGNSGD for a large class of practically realistic objectives. Therefore, we expect that these tools should carry over to help understand the *success modes* of ADAM. Our insight is that gradient noise distributions in practical problems are often unimodal and symmetric because of the Central Limit Theorem, yet Reddi et al. (2018)'s construction relies on bimodal noise distributions.

## 2 RELATED WORK

For decades, neural network researchers have adapted biologically inspired algorithms for efficient hardware implementation. Hopfield (1982), for example, considered taking the sign of the synaptic weights of his memory network for readier adaptation into integrated circuits. This past decade, neural network research has focused on training feedforward networks by gradient descent (LeCun et al., 2015). It is natural to ask what practical efficiency may accompany simply taking the sign of the backpropagated gradient. In this section, we explore related work pertaining to this question.

**Deep learning:** whilst stochastic gradient descent (SGD) is the workhorse of machine learning (Robbins & Monro, 1951), algorithms like RMSPROP (Tieleman & Hinton, 2012) and ADAM (Kingma & Ba, 2015) are also extremely popular neural net optimisers. These algorithms have their

---

**Algorithm 1** SIGNUM with majority vote, the proposed algorithm for distributed optimisation. Good default settings for the tested machine learning problems are $\eta = 0.0001$ and $\beta = 0.9$, though tuning is recommended. All operations on vectors are element-wise. Setting $\beta = 0$ yields SIGNSGD.

---

**Require:** learning rate $\eta > 0$, momentum constant $\beta \in [0, 1)$, weight decay $\lambda \geq 0$, mini-batch size $n$, initial point $x$ held by each of $M$ workers, initial momentum $v_m \leftarrow 0$ on $m^{th}$ worker

> **repeat**
>> **on** $m^{th}$ worker
>>> $\tilde{g}_m \leftarrow \frac{1}{n} \sum_{i=1}^{n} \text{stochasticGradient(x)}$       ▷ mini-batch gradient
>>> $v_m \leftarrow (1 - \beta)\tilde{g}_m + \beta v_m$          ▷ update momentum
>>> **push** $\text{sign}(v_m)$ **to** server          ▷ send sign momentum
>> **on** server
>>> $V \leftarrow \sum_{m=1}^{M} \text{sign}(v_m)$          ▷ aggregate sign momenta
>>> **push** $\text{sign}(V)$ **to** each worker        ▷ broadcast majority vote
>> **on** every worker
>>> $x \leftarrow x - \eta(\text{sign}(V) + \lambda x)$        ▷ update parameters
> **until** convergence

---

roots in the RPROP optimiser (Riedmiller & Braun, 1993), which is a sign-based method similar to SIGNSGD except for a component-wise adaptive learning rate.

**Non-convex optimisation:** parallel to (and oftentimes in isolation from) advances in deep learning practice, a sophisticated optimisation literature has developed. Nesterov & Polyak (2006) proposed *cubic regularisation* as an algorithm that can escape saddle points and provide guaranteed convergence to local minima of non-convex functions. This has been followed up by more recent works such as NATASHA (Allen-Zhu, 2017) that use other theoretical tricks to escape saddle points. It is still unclear how relevant these works are to deep learning, since it is not clear to what extent saddle points are an obstacle in practical problems. We avoid this issue altogether and satisfy ourselves with establishing convergence to critical points.

**Gradient compression:** prior work on gradient compression generally falls into two camps. In the first camp, algorithms like QSGD (Alistarh et al., 2017), TERNGRAD (Wen et al., 2017) and ATOMO (Wang et al., 2018) use stochastic quantisation schemes to ensure that the compressed stochastic gradient remains an unbiased approximation to the true gradient. These works are therefore able to bootstrap existing SGD convergence theory. In the second camp, more heuristic algorithms like 1BITSGD (Seide et al., 2014) and *deep gradient compression* (Lin et al., 2018) pay less attention to theoretical guarantees and focus more on practical performance. These algorithms track quantisation errors and feed them back into subsequent updates. The commonality between the two camps is an effort to, one way or another, correct for bias in the compression.

SIGNSGD with majority vote takes a different approach to these two existing camps. In directly employing the sign of the stochastic gradient, the algorithm unabashedly uses a biased approximation of the stochastic gradient. Carlson et al. (2016) and Bernstein et al. (2018) provide theoretical and empirical evidence that signed gradient schemes can converge well in spite of their biased nature. Their theory only applies in the large batch setting, meaning the theoretical results are less relevant to deep learning practice. Still Bernstein et al. (2018) showed promising experimental results in the mini-batch setting. An appealing feature of majority vote is that it naturally leads to compression in both directions of communication between workers and parameter server. As far as we are aware, all existing gradient compression schemes lose compression before scattering results back to workers.

**Byzantine fault tolerant optimisation:** the problem of modifying SGD to make it Byzantine fault tolerant has recently attracted interest in the literature (Yin et al., 2018). For example, Blanchard et al. (2017) proposed KRUM, which operates by detecting and excluding outliers in the gradient aggregation. Alistarh et al. (2018) propose BYZANTINESGD which instead focuses on detecting and eliminating adversaries. Clearly both these strategies incur overheads, and eliminating adversaries precludes the possibility that they might reform. El Mhamdi et al. (2018) point out that powerful adversaries may steer convergence to bad local minimisers. We see majority vote as a natural way to protect against less malign faults such as network errors, and thus satisfy ourselves with convergence guarantees to critical points without placing guarantees on their quality.

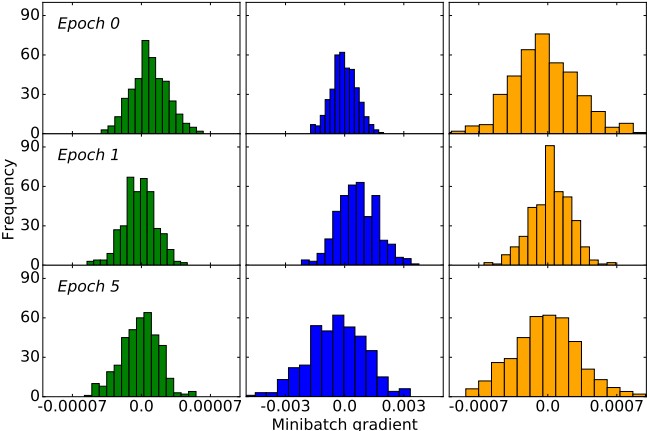

Figure 2: Gradient distributions for `resnet18` on Cifar-10 at mini-batch size 128. At the start of epochs 0, 1 and 5, we do a full pass over the data and collect the gradients for three randomly chosen weights (left, middle, right). In all cases the distribution is close to unimodal and symmetric.

## 3 THEORY

### 3.1 ASSUMPTIONS

We aim to develop an optimisation theory that is relevant for real problems in deep learning. For this reason, we are careful about the assumptions we make. For example, we do not assume convexity because neural network loss functions are typically not convex. Though we allow our objective function to be non-convex, we insist on a lower bound to enable meaningful convergence results.

**Assumption 1** (Lower bound). *For all $x$ and some constant $f^*$, we have objective value $f(x) \geq f^*$.*

Our next two assumptions of Lipschitz smoothness and bounded variance are standard in the stochastic optimisation literature (Allen-Zhu, 2017). That said, we give them in a component-wise form. This allows our convergence results to encode information not just about the total noise level and overall smoothness, but also about how these quantities are distributed across dimension.

**Assumption 2** (Smooth). *Let $g(x)$ denote the gradient of the objective $f(.)$ evaluated at point $x$. Then $\forall x, y$ we require that for some non-negative constant $\vec{L} := [L_1, ..., L_d]$*

$$\left| f(y) - \left[ f(x) + g(x)^T (y - x) \right] \right| \leq \frac{1}{2} \sum_i L_i (y_i - x_i)^2.$$

**Assumption 3** (Variance bound). *Upon receiving query $x \in \mathbb{R}^d$, the stochastic gradient oracle gives us an independent, unbiased estimate $\tilde{g}$ that has coordinate bounded variance:*

$$\mathbb{E}[\tilde{g}(x)] = g(x), \qquad \mathbb{E}\left[ (\tilde{g}(x)_i - g(x)_i)^2 \right] \leq \sigma_i^2$$

*for a vector of non-negative constants $\vec{\sigma} := [\sigma_1, .., \sigma_d]$.*

Our final assumption is non-standard. We assume that the gradient noise is unimodal and symmetric. Clearly, Gaussian noise is a special case. Note that even for a moderate mini-batch size, we expect the central limit theorem to kick in rendering typical gradient noise distributions close to Gaussian. See Figure 2 for noise distributions measured whilst training `resnet18` on Cifar-10.

**Assumption 4** (Unimodal, symmetric gradient noise). *At any given point $x$, each component of the stochastic gradient vector $\tilde{g}(x)$ has a unimodal distribution that is also symmetric about the mean.*

Showing how to work with this assumption is a key theoretical contribution of this work. Combining Assumption 4 with an old tail bound of Gauss (1823) yields Lemma 1, which will be crucial for guaranteeing mini-batch convergence of SIGNSGD. As will be explained in Section 3.3, this result also constitutes a convergence proof for a parameter regime of ADAM. This suggests that Assumption 4 may more generally be a theoretical fix for Reddi et al. (2018)'s non-convergence proof of mini-batch ADAM, a fix which does not involve modifying the ADAM algorithm itself.

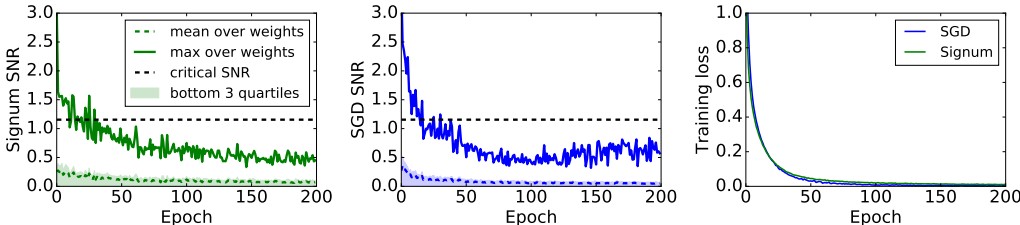

Figure 3: Signal-to-noise ratio (SNR) whilst training `resnet18` on Cifar-10 at batch size 128. At the start of each epoch we compute the SNR for every gradient component. We plot summary statistics like the mean over weights and the max. By roughly epoch 40, all gradient components have passed below the critical line (see Theorem 1) and remain there for the rest of training.

## 3.2 MINI-BATCH CONVERGENCE OF SIGNSGD

With our assumptions in place, we move on to presenting our theoretical results, which are all proved in Appendix C. Our first result establishes the mini-batch convergence behaviour of SIGNSGD. We will first state the result and make some remarks. We provide intuition for the proof in Section 3.3.

---

**Theorem 1** (Non-convex convergence rate of mini-batch SIGNSGD). *Run the following algorithm for $K$ iterations under Assumptions 1 to 4: $x_{k+1} = x_k - \eta \, \text{sign}(\tilde{g}_k)$. Set the learning rate, $\eta$, and mini-batch size, $n$, as*

$$\eta = \sqrt{\frac{f_0 - f_*}{\|\vec{L}\|_1 K}}, \qquad n = 1.$$

*Let $H_k$ be the set of gradient components at step $k$ with large signal-to-noise ratio $S_i := \frac{|g_{k,i}|}{\sigma_i}$, i.e. $H_k := \left\{ i \middle| S_i > \frac{2}{\sqrt{3}} \right\}$. We refer to $\frac{2}{\sqrt{3}}$ as the 'critical SNR'. Then we have*

$$\frac{1}{K} \sum_{k=0}^{K-1} \mathbb{E} \left[ \sum_{i \in H_k} |g_{k,i}| + \sum_{i \notin H_k} \frac{g_{k,i}^2}{\sigma_i} \right] \le 3 \sqrt{\frac{\|\vec{L}\|_1 (f_0 - f_*)}{N}}.$$

*where $N = K$ is the total number of stochastic gradient calls up to step $K$.*

---

Theorem 1 provides a bound on the average gradient norm. The right hand side of the bound decays like $\text{O}\left(\frac{1}{\sqrt{N}}\right)$, establishing convergence to points of the objective where the gradient vanishes.

**Remark 1:** mini-batch SIGNSGD attains the same $\text{O}\left(\frac{1}{\sqrt{N}}\right)$ non-convex convergence rate as SGD.

**Remark 2:** the gradient appears as a *mixed norm*: an $\ell_1$ norm for high SNR components, and a weighted $\ell_2$ norm for low SNR compoenents.

**Remark 3:** we wish to understand the dimension dependence of our bound. We may simplify matters by assuming that, during the entire course of optimisation, every gradient component lies in the low SNR regime. Figure 3 shows that this is almost true when training a `resnet18` model. In this limit, the bound becomes:

$$\frac{1}{K} \sum_{k=0}^{K-1} \mathbb{E} \left[ \sum_{i=1}^{d} \frac{g_{k,i}^2}{\sigma_i} \right] \le 3 \sqrt{\frac{\|\vec{L}\|_1 (f_0 - f_*)}{N}}.$$

Further assume that we are in a *well-conditioned* setting, meaning that the variance is distributed uniformly across dimension ($\sigma_i^2 = \frac{\sigma^2}{d}$), and every weight has the same smoothness constant ($L_i = L$). $\sigma^2$ is the total variance bound, and $L$ is the conventional Lipschitz smoothness. These are the

quantities which appear in the standard analysis of SGD. Then we get

$$\frac{1}{K}\sum_{k=0}^{K-1}\mathbb{E}\|g_k\|_2^2 \le 3\sigma\sqrt{\frac{L(f_0 - f_*)}{N}}.$$

The factors of dimension $d$ have conveniently cancelled. This illustrates that there are problem geometries where mini-batch SIGNSGD does not pick up an unfavourable dimension dependence.

### 3.3 THE SUBTLETIES OF MINI-BATCH CONVERGENCE

Intuitively, the convergence analysis of SIGNSGD depends on the probability that a given bit of the sign stochastic gradient vector is incorrect, or $\mathbb{P}[\text{sign}(\tilde{g}_i) \ne \text{sign}(g_i)]$. Lemma 1 provides a bound on this quantity under Assumption 4 (unimodal symmetric gradient noise).

---

**Lemma 1** (Bernstein et al. (2018)). *Let $\tilde{g}_i$ be an unbiased stochastic approximation to gradient component $g_i$, with variance bounded by $\sigma_i^2$. Further assume that the noise distribution is unimodal and symmetric. Define signal-to-noise ratio $S_i := \frac{|g_i|}{\sigma_i}$. Then we have that*

$$\mathbb{P}[\text{sign}(\tilde{g}_i) \ne \text{sign}(g_i)] \le \begin{cases} \frac{2}{9}\frac{1}{S_i^2} & \text{if } S_i > \frac{2}{\sqrt{3}}, \\ \frac{1}{2} - \frac{S_i}{2\sqrt{3}} & \text{otherwise} \end{cases}$$

*which is in all cases less than or equal to $\frac{1}{2}$.*

---

The bound characterises how the failure probability of a sign bit depends on the *signal-to-noise* ratio (SNR) of that gradient component. Intuitively as the SNR decreases, the quality of the sign estimate should degrade. The bound is important since it tells us that, under conditions of unimodal symmetric gradient noise, even at extremely low SNR we still have that $\mathbb{P}[\text{sign}(\tilde{g}_i) \ne \text{sign}(g_i)] \le \frac{1}{2}$. This means that even when the gradient is very small compared to the noise, the sign stochastic gradient still tells us, on average, useful information about the true gradient direction, allowing us to guarantee convergence as in Theorem 1.

Without Assumption 4, the mini-batch algorithm may not converge. This is best appreciated with a simple example. Consider a stochastic gradient component $\tilde{g}$ with bimodal noise:

$$\tilde{g} = \begin{cases} 50 & \text{with probability } 0.1; \\ -1 & \text{with probability } 0.9. \end{cases}$$

The true gradient $g = \mathbb{E}[\tilde{g}] = 4.1$ is positive. But the sign gradient $\text{sign}(\tilde{g})$ is negative with probability 0.9. Therefore SIGNSGD will tend to move in the wrong direction for this noise distribution.

Note that SIGNSGD is a special case of the ADAM algorithm (Balles & Hennig, 2018). To see this, set $\beta_1 = \beta_2 = \epsilon = 0$ in ADAM, and the ADAM update becomes:

$$-\frac{\tilde{g}}{\sqrt{\tilde{g}^2}} = -\frac{\tilde{g}}{|\tilde{g}|} = -\text{sign}(\tilde{g})$$

This correspondence suggests that Assumption 4 should be useful for obtaining mini-batch convergence guarantees for ADAM. Note that when Reddi et al. (2018) construct toy divergence examples for ADAM, they rely on bimodal noise distributions which violate Assumption 4.

We conclude this section by noting that without Assumption 4, SIGNSGD can still be guaranteed to converge. The trick is to use a "large" batch size that grows with the number of iterations. This will ensure that the algorithm stays in the high SNR regime where the failure probability of the sign bit is low. This is the approach taken by both Carlson et al. (2016) and Bernstein et al. (2018).

### 3.4 ROBUSTNESS OF CONVERGENCE

We will now study SIGNSGD's robustness when distributed by majority vote. We model adversaries as machines that may manipulate their stochastic gradient as follows.

**Definition 1** (Blind multiplicative adversary). *A blind multiplicative adversary may manipulate their stochastic gradient estimate $\tilde{g}_t$ at iteration $t$ by element-wise multiplying $\tilde{g}_t$ with any vector $v_t$ of their choice. The vector $v_t$ must be chosen before observing $\tilde{g}_t$, so the adversary is 'blind'. Some interesting members of this class are:*

  *(i) adversaries that arbitrarily rescale their stochastic gradient estimate;*

  *(ii) adversaries that randomise the sign of each coordinate of the stochastic gradient;*

  *(iii) adversaries that invert their stochastic gradient estimate.*

SGD is certainly not robust to rescaling since an adversary could set the gradient to infinity and corrupt the entire model. Our algorithm, on the other hand, is robust to all adversaries in this class. For ease of analysis, here we derive large batch results. We make sure to give results in terms of sample complexity $N$ (and not iteration number $K$) to enable fair comparison with other algorithms.

---

**Theorem 2** (Non-convex convergence rate of majority vote with adversarial workers). *Run algorithm 1 for $K$ iterations under Assumptions 1 to 4. Switch off momentum and weight decay ($\beta = \lambda = 0$). Set the learning rate, $\eta$, and mini-batch size, $n$, for each worker as*

$$\eta = \sqrt{\frac{f_0 - f_*}{\|L\|_1 K}}, \qquad n = K.$$

*Assume that a fraction $\alpha < \frac{1}{2}$ of the $M$ workers behave adversarially according to Definition 1. Then majority vote converges at rate:*

$$\left[ \frac{1}{K} \sum_{k=0}^{K-1} \mathbb{E} \|g_k\|_1 \right]^2 \leq \frac{4}{\sqrt{N}} \left[ \frac{1}{1-2\alpha} \frac{\|\vec{\sigma}\|_1}{\sqrt{M}} + \sqrt{\|L\|_1 (f_0 - f^*)} \right]^2$$

*where $N = K^2$ is the total number of stochastic gradient calls per worker up to step $K$.*

---

The result is intuitive: provided there are more machines sending honest gradients than adversarial gradients, we expect that the majority vote should come out correct on average.

**Remark 1:** if we switch off adversaries by setting the proportion of adversaries $\alpha = 0$, this result reduces to Theorem 2 in (Bernstein et al., 2018). In this case, we note the $\frac{1}{\sqrt{M}}$ variance reduction that majority vote obtains by distributing over $M$ machines, similar to distributed SGD.

**Remark 2:** the convergence rate degrades as we ramp up $\alpha$ from 0 to $\frac{1}{2}$.

**Remark 3:** from an optimisation theory perspective, the large batch size is an advantage. This is because when using a large batch size, fewer iterations and rounds of communication are theoretically needed to reach a desired accuracy, since only $\sqrt{N}$ iterations are needed to reach $N$ samples. But from a practical perspective, workers may be unable to handle such a large batch size in a timely manner. It should be possible to extend the result to the mini-batch setting by combining the techniques of Theorems 1 and 2, but we leave this for future work.

## 4 EXPERIMENTS

For our experiments, we distributed SIGNUM (Algorithm 1) by majority vote. SIGNUM is the momentum counterpart of SIGNSGD, where each worker maintains a momentum and transmits the sign momentum to the parameter server at each step. The addition of momentum to SIGNSGD is proposed and studied in (Balles & Hennig, 2018; Bernstein et al., 2018).

We built SIGNUM with majority vote in the Pytorch deep learning framework (Paszke et al., 2017) using the Gloo (2018) communication library. Unfortunately Pytorch and Gloo do not natively support 1-bit tensors, therefore we wrote our own compression code to bit-pack a sign tensor down to an efficient 1-bit representation. We obtained a performance boost by fusing together smaller tensors, which saved on compression and communication costs.

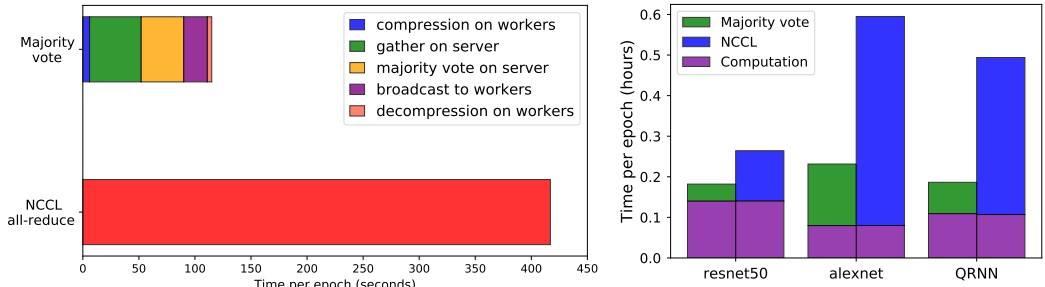

Figure 4: Timing breakdown for distributing on the cloud. Left: comparing communication (including compression) for training `resnet50`. Right: comparing communication (including compression) and computation. `resnet50` results use 7 `p3.2xlarge` machines for training Imagenet, each at batch size 128. `alexnet` uses 7 `p3.2xlarge` machines for Imagenet, each at batch size 64. QRNN uses 3 `p3.16xlarge` machines for training `WikiText-103`, each at batch size 240.

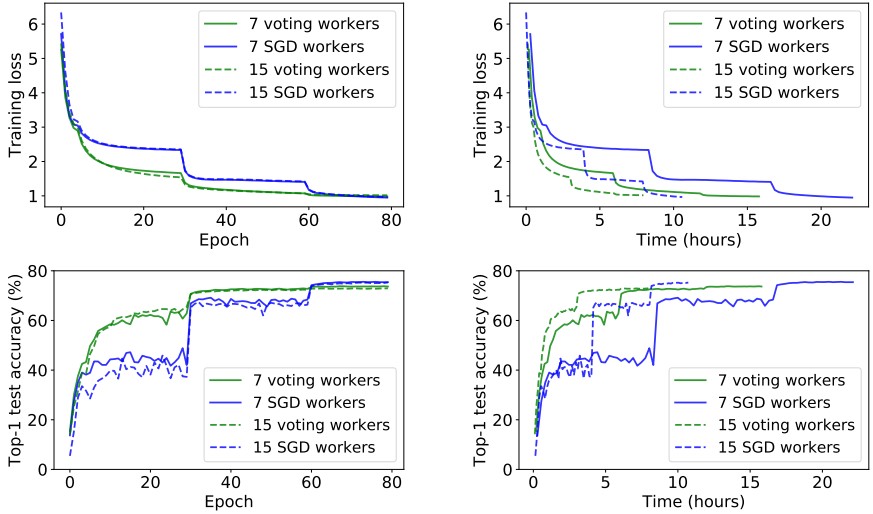

Figure 5: Imagenet comparison of SIGNUM with majority vote and SGD distributed with NCCL. We train `resnet50` on Imagenet distributed over 7 to 15 AWS `p3.2xlarge` machines. Top: increasing the number of workers participating in the majority vote shows a similar convergence speedup to distributed SGD. But in terms of wall-clock time, majority vote training is roughly 25% faster for the same number of epochs. Bottom: in terms of generalisation accuracy, majority vote shows a slight degradation compared to SGD. Perhaps a better regularisation scheme can fix this.

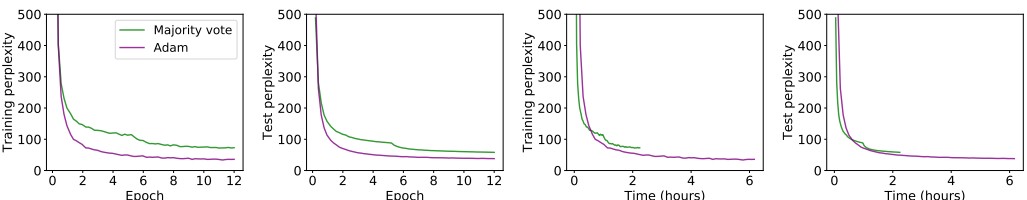

Figure 6: Training QRNN across three `p3.16xlarge` machines on `WikiText-103`. Each machine uses a batch size of 240. For ADAM, the gradient is aggregated with NCCL. SIGNUM with majority vote shows some degradation compared to ADAM, although an epoch is completed roughly three times faster. This means that after 2 hours of training, SIGNUM attains a similar perplexity to ADAM. Increasing the per-worker batch size improved SIGNUM's performance (see Appendix A), and increasing it beyond 240 may further improve SIGNUM's performance. Note: the test perplexity beats training perplexity because dropout was applied during training but not testing.

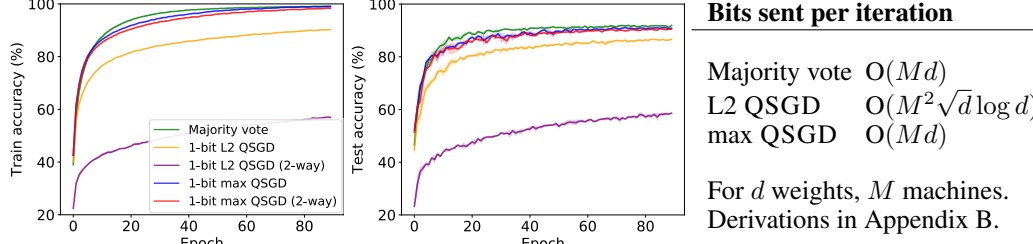

Figure 7: Left: comparing convergence of majority vote to QSGD (Alistarh et al., 2017). `resnet18` is trained on Cifar-10 across $M = 3$ machines, each at bach size 128. 1-bit QSGD stochastically snaps gradient components to $\{0, \pm 1\}$. 2-way refers to the compression function $Q(.)$ being applied in both directions of communication: machine $i$ sends $Q(\tilde{g}_i)$ to the server and gets $Q(\sum_{i=1}^{M} Q(\tilde{g}_i))$ sent back. Alistarh et al. (2017) develop a theory for *L2* QSGD, but experimentally benchmark *max* QSGD which has much larger communication costs. For this experiment, 1-bit max QSGD gives roughly $5\times$ more compression than the $32\times$ compression of majority vote, but this further gain turns out to be small relative to the cost of backpropagation. See Appendix A for QSGD experiments at higher bit-precision. Right: the table gives a theoretical comparison of the compression cost of each algorithm—see Appendix B for derivations.

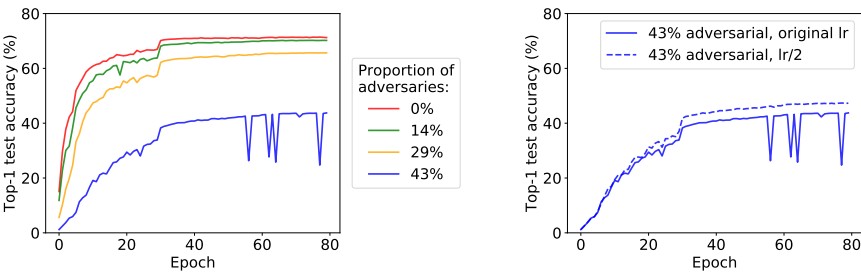

Figure 8: Imagenet robustness experiments. We used majority vote to train `resnet50` distributed across 7 AWS `p3.2xlarge` machines. Adversaries invert their sign stochastic gradient. Left: all experiments are run at identical hyperparameter settings, with weight decay switched off for simplicity. The network still learns even at 43% adversarial. Right: at 43% adversarial, learning became slightly unstable. We decreased the learning rate for this setting, and learning stabilised.

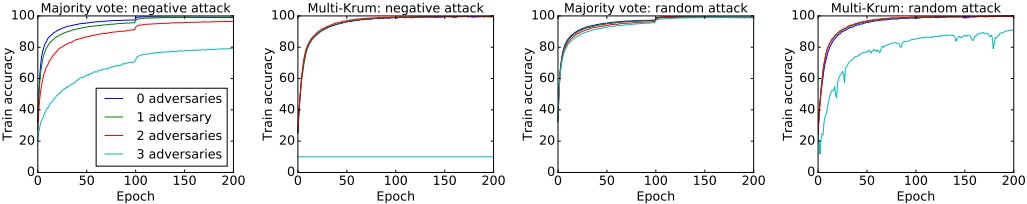

Figure 9: Comparing the robustness of majority vote to MULTI-KRUM (Blanchard et al., 2017). We train `resnet18` on Cifar-10 across 7 workers, each at batch size 64. Momentum and weight decay are switched off for simplicity, and for majority vote we divide the learning rate by 10 at epoch 100. Negative adversaries multiply their stochastic gradient estimate by $-10$. Random adversaries multiply their stochastic gradient estimate by 10 and then randomise the sign of each coordinate. For MULTI-KRUM, we use the maximum allowed security level of $f = 2$. Notice that MULTI-KRUM fails catastrophically once the number of adversaries exceeds the security level, whereas majority vote fails more gracefully.

We test against SGD distributed using the state of the art NCCL (2018) communication library. NCCL provides an efficient implementation of allreduce. Our framework is often $4\times$ faster in communication (including the cost of compression) than NCCL, as can be seen in Figure 4. Further code optimisation should bring the speedup closer to the ideal $32\times$.

## 4.1 COMMUNICATION EFFICIENCY

We first benchmark majority vote on the Imagenet dataset. We train a `resnet50` model and disitribute learning over 7 to 15 AWS `p3.2xlarge` machines. These machines each contain one Nvidia Tesla V100 GPU, and AWS lists the connection speed between machines as "up to 10 Gbps". Results are plotted in Figure 5. *Per epoch*, distributing by majority vote is able to attain a similar speedup to distributed SGD. But *per hour* majority vote is able to process more epochs than NCCL, meaning it can complete the 80 epoch training job roughly 25% faster. In terms of overall generalisation, majority vote reaches a slightly degraded test set accuracy. We hypothesise that this may be fixed by inventing a better regularisation scheme or tuning momentum, which we did not do.

In Figure 6 we compare majority vote to ADAM (distributed by NCCL) for training QRNN (Bradbury et al., 2017) on `WikiText-103`. Majority vote completes an epoch roughly 3 times faster than ADAM, but it reaches a degraded accuracy so that the overall test perplexity after 2 hours ends up being similar. In Figure 7 we show that majority vote has superior convergence to the 'theory' version of QSGD that Alistarh et al. (2017) develop. Convergence is similar for the 'max' version that Alistarh et al. (2017) use in their experiments. Additional results are given in Appendix A.

## 4.2 ROBUSTNESS

In this section we test the robustness of SIGNUM with majority vote to Byzantine faults. Again we run tests on the Imagenet dataset, training `resnet50` across 7 AWS `p3.2xlarge` machines. Our adversarial workers take the sign of their stochastic gradient calculation, but send the negation to the parameter server. Our results are plotted in Figure 8. In the left hand plot, all experiments were carried out using hyperparameters tuned for the 0% adversarial case. Weight decay was not used in these experiments to simplify matters. We see that learning is tolerant of up to 43% (3 out of 7) machines behaving adversarially. The 43% adversarial case was slightly unstable (Figure 8, left), but re-tuning the learning rate for this specific case stabilised learning (Figure 8, right).

In Figure 9 we compare majority vote to MULTI-KRUM (Blanchard et al., 2017) with a security level of $f = 2$. When the number of adversaries exceeds $f$, MULTI-KRUM fails catastrophically in our experiments, whereas SIGNSGD fails more gracefully. Note that MULTI-KRUM requires $2f + 2 < M$, therefore $f = 2$ is the maximum possible security level for these experiments with $M = 7$ workers.

## 5 DISCUSSION AND CONCLUSION

We have analysed the theoretical and empirical properties of a very simple algorithm for distributed, stochastic optimisation. We have shown that SIGNSGD with majority vote aggregation is robust and communication efficient, whilst its per-iteration convergence rate is competitive with SGD for training large-scale convolutional neural nets on image datasets. We believe that it is important to understand this simple algorithm before going on to devise more complex learning algorithms.

An important takeaway from our theory is that mini-batch SIGNSGD should converge if the gradient noise is Gaussian. This means that the performance of SIGNSGD may be improved by increasing the per-worker mini-batch size, since this should make the noise 'more Gaussian' according to the Central Limit Theorem.

We will now give some possible directions for future work. Our implementation of majority vote may be further optimised by breaking up the parameter server and distributing it across machines. This would prevent a single machine from becoming a communication bottleneck as in our experiments. Though our framework speeds up Imagenet training, we still have a test set gap. Future work could attempt to devise new regularisation schemes for signed updates to close this gap. Promising future work could also explore the link between SIGNSGD and model compression. Signed updates force the weights to live on a lattice, facilitating compression of the resulting model.

ACKNOWLEDGMENTS

We would like to thank Yu-Xiang Wang, Alexander Sergeev, Soumith Chintala, Pieter Noordhuis, Hongyi Wang, Scott Sievert and El Mahdi El Mhamdi for useful discussions.

KA is supported in part by NSF Career Award CCF-1254106. AA is supported in part by a Microsoft Faculty Fellowship, Google Faculty Award, Adobe Grant, NSF Career Award CCF-1254106, and AFOSR YIP FA9550-15-1-0221.

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

# A  ADDITIONAL EXPERIMENTS

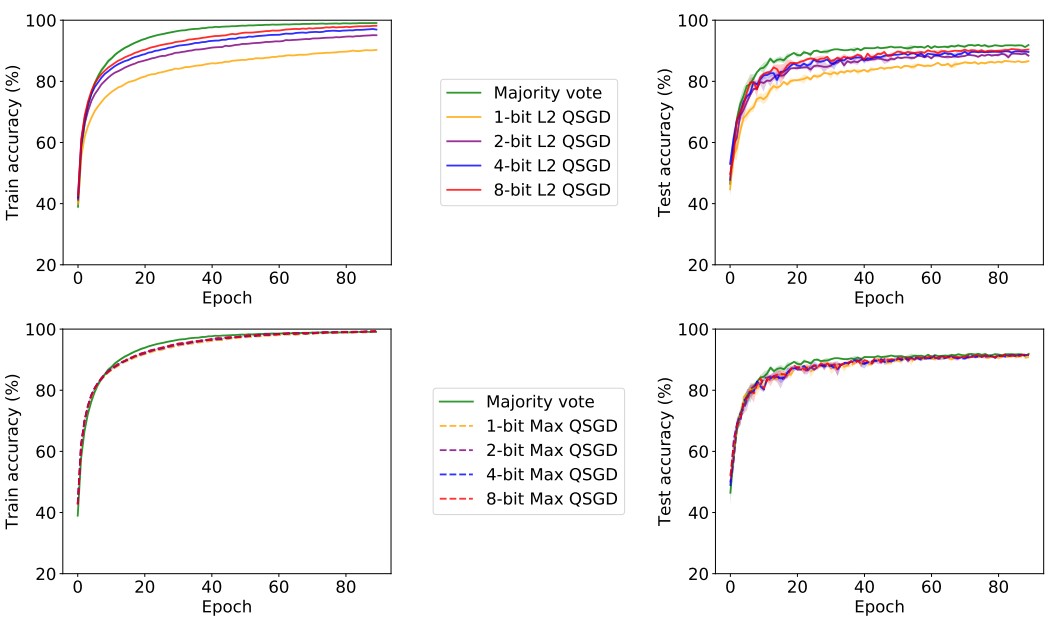

Figure 10: QSGD at varying levels of precision. Top row: L2 QSGD. Bottom row: max QSGD. resnet18 is trained on Cifar-10 across $M = 3$ machines, each at bach size 128. The one-way version of QSGD is used, meaning that the compression function is not re-applied after aggregation. The comparison is given in terms of number of epochs. A comparison in terms of wall-clock time will depend on details of the systems implementation.

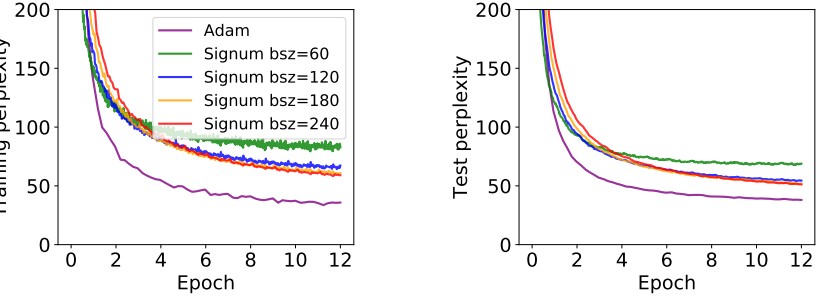

Figure 11: SIGNUM at varying batch sizes. We use a single worker and train a QRNN model on WikiText-103. ADAM is shown for comparison, at batch size 60. The performance of SIGNUM is seen to improve with increasing batch size.

## B   BITS SENT PER ITERATION: SIGNSGD VS. QSGD

In this section, we perform theoretical calculations of the number of bits sent per iteration in distributed training. We compare SIGNSGD using majority vote aggregation to two forms of QSGD (Alistarh et al., 2017). These calculations give the numbers in the table in Figure 7.

The communication cost of SIGNSGD with majority vote is trivially $2Md$ bits per iterations, since at each iteration $M$ machines send $d$-dimensional sign vectors up to the server, and the server sends back one $d$-dimensional sign vector to all $M$ machines.

There are two variants of QSGD given in (Alistarh et al., 2017). The first we refer to as *L2* QSGD which is the version developed in the theory section of (Alistarh et al., 2017). The second we refer to as *max* QSGD which is the version actually used in their experiments. For each version we compute the number of bits sent for the highest compression version of the algorithm, which is a ternary quantisation (snapping gradient components into $\{0, \pm 1\}$). We refer to this as *1-bit* QSGD. The higher precision versions of QSGD will send more bits per iteration.

1-bit L2 QSGD takes a gradient vector $g$ and snaps $i^{th}$ coordinate $g_i$ to $\text{sign}(g_i)$ with probability $\frac{|g_i|}{\|g\|_2}$ and sets it to zero otherwise. Therefore the expected number of bits set to $\pm 1$ is bounded by

$$\mathbb{E}[\#\text{bits}] = \sum_{i=1} \frac{|g_i|}{\|g\|_2} = \frac{\|g\|_1}{\|g\|_2} \leq \sqrt{d}.$$

To send a vector compressed in this way, for each non-zero component 1 bit is needed to send the sign and $\log d$ bits are needed to send the index. Therefore sending a vector compressed by 1-bit L2 QSGD requires at most $\sqrt{d}(1 + \log d)$ bits.

In the experiments in Figure 7 we see that the '2-way' version of 1-bit L2 QSGD (which recompresses the aggregated compressed gradients) converges very poorly. Therefore it makes sense to use the 1-way version where the aggregated compressed gradient is not recompressed. A sensible way to enact this is to have each of the $M$ workers broadcast their compressed gradient vector to all other workers. This has a cost of $(M - 1)\sqrt{d}(1 + \log d)$ bits for each of the $M$ workers, and from this we get the total cost of $\text{O}(M^2 \sqrt{d} \log d)$ for 1-bit L2 QSGD.

The final algorithm to characterise is 1-bit max QSGD. 1-bit max QSGD takes a gradient vector $g$ and snaps $i^{th}$ coordinate $g_i$ to $\text{sign}(g_i)$ with probability $\frac{|g_i|}{\|g\|_\infty}$ and sets it to zero otherwise. As noted in (Alistarh et al., 2017), there are no sparsity guarantees for this algorithm, so compression will generally be much lower than for 1-bit L2 QSGD.

It is easy to see that 1-bit max QSGD requires no more than $\text{O}(d)$ bits to compress a $d$-dimensional vector, since $2d$ bits can always store $d$ numbers in $\{0, \pm 1\}$. To see that we can't generally do better than $\text{O}(d)$ bits, notice that 1-bit max QSGD leaves sign vectors invariant, and thus the compressed form of a sign vector requires exactly $d$ bits. The natural way to enact 1-bit max QSGD is with a two-way compression where the $M$ workers each send an $\text{O}(d)$-bit compressed gradient up to the server, and the server sends back an $\text{O}(d)$-bit compressed aggregated result back to the $M$ workers. This gives a number of bits sent per iteration of $\text{O}(Md)$.

For very sparse vectors 1-bit max QSGD will compress much better than indicated above. For a vector $g$ with a single non-zero entry, 1-bit max QSGD will set this entry to 1 and keep the rest zero, thus requiring only $\log d$ bits to send the index of the non-zero entry. But it is not clear whether these extremely sparse vectors appear in deep learning problems. In the experiments in Figure 7, 1-bit max QSGD led to compressed vectors that were $5\times$ more compressed than SIGNSGD—in our experimental setting this additional improvement turned out to be small relative to the time cost of backpropagation.

# C  PROOFS

## C.1  ACCURACY OF THE SIGN STOCHASTIC GRADIENT

**Lemma 1** (Bernstein et al. (2018)). *Let $\tilde{g}_i$ be an unbiased stochastic approximation to gradient component $g_i$, with variance bounded by $\sigma_i^2$. Further assume that the noise distribution is unimodal and symmetric. Define signal-to-noise ratio $S_i := \frac{|g_i|}{\sigma_i}$. Then we have that*

$$\mathbb{P}[\mathrm{sign}(\tilde{g}_i) \neq \mathrm{sign}(g_i)] \leq \begin{cases} \frac{2}{9}\frac{1}{S_i^2} & \text{if } S_i > \frac{2}{\sqrt{3}}, \\ \frac{1}{2} - \frac{S_i}{2\sqrt{3}} & \text{otherwise} \end{cases}$$

*which is in all cases less than or equal to $\frac{1}{2}$.*

*Proof.* Recall Gauss' inequality for unimodal random variable X with mode $\nu$ and expected squared deviation from the mode $\tau^2$ (Gauss, 1823; Pukelsheim, 1994):

$$\mathbb{P}[|X - \nu| > k] \leq \begin{cases} \frac{4}{9}\frac{\tau^2}{k^2} & \text{if } \frac{k}{\tau} > \frac{2}{\sqrt{3}}, \\ 1 - \frac{k}{\sqrt{3}\tau} & \text{otherwise} \end{cases}$$

By the symmetry assumption, the mode is equal to the mean, so we replace mean $\mu = \nu$ and variance $\sigma^2 = \tau^2$.

$$\mathbb{P}[|X - \mu| > k] \leq \begin{cases} \frac{4}{9}\frac{\sigma^2}{k^2} & \text{if } \frac{k}{\sigma} > \frac{2}{\sqrt{3}}, \\ 1 - \frac{k}{\sqrt{3}\sigma} & \text{otherwise} \end{cases}$$

Without loss of generality assume that $g_i$ is negative. Then applying symmetry followed by Gauss, the failure probability for the sign bit satisfies:

$$\begin{aligned}
\mathbb{P}[\mathrm{sign}(\tilde{g}_i) \neq \mathrm{sign}(g_i)] &= \mathbb{P}[\tilde{g}_i - g_i \geq |g_i|] \\
&= \frac{1}{2}\mathbb{P}[|\tilde{g}_i - g_i| \geq |g_i|] \\
&\leq \begin{cases} \frac{2}{9}\frac{\sigma_i^2}{g_i^2} & \text{if } \frac{|g_i|}{\sigma} > \frac{2}{\sqrt{3}}, \\ \frac{1}{2} - \frac{|g_i|}{2\sqrt{3}\sigma_i} & \text{otherwise} \end{cases} \\
&= \begin{cases} \frac{2}{9}\frac{1}{S_i^2} & \text{if } S_i > \frac{2}{\sqrt{3}}, \\ \frac{1}{2} - \frac{S_i}{2\sqrt{3}} & \text{otherwise} \end{cases}
\end{aligned}$$

$\square$

## C.2  MINI-BATCH CONVERGENCE GUARANTEES

**Theorem 1** (Non-convex convergence rate of mini-batch SIGNSGD). *Run the following algorithm for $K$ iterations under Assumptions 1 to 4: $x_{k+1} = x_k - \eta\,\mathrm{sign}(\tilde{g}_k)$. Set the learning rate, $\eta$, and mini-batch size, $n$, as*

$$\eta = \sqrt{\frac{f_0 - f_*}{\|\vec{L}\|_1 K}}, \qquad n = 1.$$

*Let $H_k$ be the set of gradient components at step $k$ with large signal-to-noise ratio $S_i := \frac{|g_{k,i}|}{\sigma_i}$, i.e. $H_k := \left\{ i \,\middle|\, S_i > \frac{2}{\sqrt{3}} \right\}$. We refer to $\frac{2}{\sqrt{3}}$ as the 'critical SNR'. Then we have*

$$\frac{1}{K}\sum_{k=0}^{K-1}\mathbb{E}\left[\sum_{i \in H_k}|g_{k,i}| + \sum_{i \notin H_k}\frac{g_{k,i}^2}{\sigma_i}\right] \leq 3\sqrt{\frac{\|\vec{L}\|_1(f_0 - f_*)}{N}}.$$

*where $N = K$ is the total number of stochastic gradient calls up to step $K$.*

*Proof.* First let's bound the improvement of the objective during a single step of the algorithm for one instantiation of the noise. $\mathbb{I}[.]$ is the indicator function, $g_{k,i}$ denotes the $i^{th}$ component of the true gradient $g(x_k)$ and $\tilde{g}_k$ is a stochastic sample obeying Assumption 3.

First take Assumption 2, plug in the algorithmic step, and decompose the improvement to expose the stochasticity-induced error:

$$f_{k+1} - f_k \le g_k^T(x_{k+1} - x_k) + \sum_{i=1}^d \frac{L_i}{2}(x_{k+1} - x_k)_i^2$$

$$= -\eta g_k^T \operatorname{sign}(\tilde{g}_k) + \eta^2 \sum_{i=1}^d \frac{L_i}{2}$$

$$= -\eta\|g_k\|_1 + \frac{\eta^2}{2}\|\vec{L}\|_1 + 2\eta\sum_{i=1}^d |g_{k,i}|\,\mathbb{I}[\operatorname{sign}(\tilde{g}_{k,i}) \ne \operatorname{sign}(g_{k,i})]$$

Next we find the expected improvement at time $k+1$ conditioned on the previous iterate.

$$\mathbb{E}[f_{k+1} - f_k|x_k] \le -\eta\|g_k\|_1 + \frac{\eta^2}{2}\|\vec{L}\|_1 + 2\eta\sum_{i=1}^d |g_{k,i}|\,\mathbb{P}[\operatorname{sign}(\tilde{g}_{k,i}) \ne \operatorname{sign}(g_{k,i})]$$

By Assumption 4 and Lemma 1 we have the following bound on the failure probability of the sign:

$$\mathbb{P}[\operatorname{sign}(\tilde{g}_i) \ne \operatorname{sign}(g_i)] \le \begin{cases} \frac{2}{9}\frac{1}{S_i^2} & \text{if } S_i > \frac{2}{\sqrt{3}}, \\ \frac{1}{2} - \frac{S_i}{2\sqrt{3}} & \text{otherwise} \end{cases}$$

$$\le \begin{cases} \frac{1}{6} & \text{if } S_i > \frac{2}{\sqrt{3}}, \\ \frac{1}{2} - \frac{S_i}{2\sqrt{3}} & \text{otherwise} \end{cases}$$

Substituting this in, we get that

$$\mathbb{E}[f_{k+1} - f_k|x_k] \le -\eta\|g_k\|_1 + \frac{\eta^2}{2}\|\vec{L}\|_1 + 2\eta\sum_{i\in H_k}\frac{|g_{k,i}|}{6} + 2\eta\sum_{i\notin H_k}|g_{k,i}|\left[\frac{1}{2} - \frac{|g_{k,i}|}{2\sqrt{3}\sigma_i}\right]$$

$$= -\eta\sum_{i=1}^d |g_{k,i}| + \frac{\eta^2}{2}\|\vec{L}\|_1 + \eta\sum_{i\in H_k}\frac{|g_{k,i}|}{3} + \eta\sum_{i\notin H_k}|g_{k,i}| - \eta\sum_{i\notin H_k}\frac{g_{k,i}^2}{\sqrt{3}\sigma_i}$$

$$= -\frac{2\eta}{3}\sum_{i\in H_k}|g_{k,i}| - \eta\sum_{i\notin H_k}\frac{g_{k,i}^2}{\sqrt{3}\sigma_i} + \frac{\eta^2}{2}\|\vec{L}\|_1$$

Interestingly a mixture between an $\ell_1$ and a variance weighted $\ell_2$ norm has appeared. Now substitute in the learning rate schedule, and we get:

$$\mathbb{E}[f_{k+1} - f_k|x_k] \le -\sqrt{\frac{f_0 - f_*}{\|\vec{L}\|_1 K}}\left[\frac{2}{3}\sum_{i\in H_k}|g_{k,i}| + \frac{1}{\sqrt{3}}\sum_{i\notin H_k}\frac{g_{k,i}^2}{\sigma_i}\right] + \frac{f_0 - f_*}{2K}$$

$$\le -\sqrt{\frac{f_0 - f_*}{3\|\vec{L}\|_1 K}}\left[\sum_{i\in H_k}|g_{k,i}| + \sum_{i\notin H_k}\frac{g_{k,i}^2}{\sigma_i}\right] + \frac{f_0 - f_*}{2K}$$

Now extend the expectation over the randomness in the trajectory and telescope over the iterations:

$$f_0 - f^* \ge f_0 - \mathbb{E}[f_K]$$

$$= \mathbb{E}\left[\sum_{k=0}^{K-1} f_k - f_{k+1}\right]$$

$$\ge \sqrt{\frac{f_0 - f_*}{3\|\vec{L}\|_1 K}}\sum_{k=0}^{K-1}\mathbb{E}\left[\sum_{i\in H_k}|g_{k,i}| + \sum_{i\notin H_k}\frac{g_{k,i}^2}{\sigma_i}\right] - \frac{f_0 - f_*}{2}$$

Finally, rearrange and substitute in $N = K$ to yield the bound

$$\frac{1}{K} \sum_{k=0}^{K-1} \mathbb{E} \left[ \sum_{i \in H_k} |g_{k,i}| + \sum_{i \notin H_k} \frac{g_{k,i}^2}{\sigma_i} \right] \leq \frac{3\sqrt{3}}{2} \sqrt{\frac{\|\vec{L}\|_1 (f_0 - f_*)}{N}} \leq 3 \sqrt{\frac{\|\vec{L}\|_1 (f_0 - f_*)}{N}}.$$

$\square$

### C.3 ROBUSTNESS OF MAJORITY VOTE

**Theorem 2** (Non-convex convergence rate of majority vote with adversarial workers). *Run algorithm 1 for $K$ iterations under Assumptions 1 to 4. Switch off momentum and weight decay ($\beta = \lambda = 0$). Set the learning rate, $\eta$, and mini-batch size, $n$, for each worker as*

$$\eta = \sqrt{\frac{f_0 - f_*}{\|L\|_1 K}}, \qquad n = K.$$

*Assume that a fraction $\alpha < \frac{1}{2}$ of the $M$ workers behave adversarially according to Definition 1. Then majority vote converges at rate:*

$$\left[ \frac{1}{K} \sum_{k=0}^{K-1} \mathbb{E} \|g_k\|_1 \right]^2 \leq \frac{4}{\sqrt{N}} \left[ \frac{1}{1 - 2\alpha} \frac{\|\vec{\sigma}\|_1}{\sqrt{M}} + \sqrt{\|L\|_1 (f_0 - f^*)} \right]^2$$

*where $N = K^2$ is the total number of stochastic gradient calls per worker up to step $K$.*

*Proof.* We need to bound the failure probability of the vote. We can then use this bound to derive a convergence rate. We will begin by showing this bound is worst when the adversary inverts the signs of the sign stochastic gradient.

Given an adversary from the class of blind multiplicative adversaries (Definition 1), the adversary may manipulate their stochastic gradient estimate $\tilde{g}_t$ into the form $v_t \otimes \tilde{g}_t$. Here $v_t$ is a vector of the adversary's choosing, and $\otimes$ denotes element-wise multiplication. The sign of this quantity obeys:

$$\text{sign}(v_t \otimes \tilde{g}_t) = \text{sign}(v_t) \otimes \text{sign}(\tilde{g}_t).$$

Therefore, the only thing that matters is the sign of $v_t$, and rescaling attacks are immediately nullified. For each component of the stochastic gradient, the adversary must decide (without observing $g_t$, since the adversary is blind) whether or not they would like to invert the sign of that component. We will now show that the failure probability of the vote is always larger when the adversary decides to invert (by setting every component of $\text{sign}(v_t)$ to $-1$). Our analysis will then proceed under this worst case.

For a gradient component with true value $g$, let random variable $Z \in [0, M]$ denote the number of correct sign bits received by the parameter server. For a given adversary, we may decompose $Z$ into the contribution from that adversary and a residual term $X$ from the remaining workers (both regular and adversarial):

$$Z(\text{sign}(v)) = X + \mathbb{I}[\text{sign}(v)\text{sign}(\tilde{g}) = \text{sign}(g)],$$

where $\tilde{g}$ is the adversary's stochastic gradient estimate for that component, $v$ is the adversary's chosen scalar for that component, and $\mathbb{I}$ is the 0-1 indicator function. We are considering Z to be a function of $\text{sign}(v)$.

But by Assumption 4 and Lemma 1, we see that $\mathbb{I}[+1 \times \text{sign}(\tilde{g}) = \text{sign}(g)]$ is a Bernoulli random variable with success probability $p \geq \frac{1}{2}$. On the other hand, $\mathbb{I}[-1 \times \text{sign}(\tilde{g}) = \text{sign}(g)]$ is a Bernoulli random variable with success probability $q = 1 - p \leq \frac{1}{2}$.

The essential quantity in our analysis is the probability that more than half the workers provide the correct sign bit. But from the preceding discussion, this clearly obeys:

$$\mathbb{P}\left[Z(+1) \leq \frac{M}{2}\right] = \mathbb{P}\left[X + \mathbb{I}[+1 \times \text{sign}(\tilde{g}) = \text{sign}(g)] \leq \frac{M}{2}\right]$$

$$\leq \mathbb{P}\left[X + \mathbb{I}[-1 \times \text{sign}(\tilde{g}) = \text{sign}(g)] \leq \frac{M}{2}\right]$$

$$= \mathbb{P}\left[Z(-1) \leq \frac{M}{2}\right].$$

As we will see below, the implication of this is that our bounds always worse under the setting $v = -1$, and so we will adopt $v = -1$ hereon. It is worth remarking that blindness in the definition of blind multiplicative adversaries is important to ensure that $\mathbb{I}[\text{sign}(v)\text{sign}(\tilde{g}) = \text{sign}(g)]$ is indeed a random variable as described above. Were the adversary not blind, then the adversary could in effect deterministically set $\text{sign}(v)\text{sign}(\tilde{g})$. Cooperative adversaries, for example, could use this power to control the vote.

Now we restrict to our worst case blind multiplicative adversaries, that always choose to invert their sign stochastic gradient estimate. So, we have $(1 - \alpha)M$ good machines and $\alpha M$ adversaries. The good workers each compute a stochastic gradient estimate, take its sign and transmit this to the server. The bad workers follow an identical procedure except they negate their sign bits prior to transmission to the server. It is intuitive that because the proportion of adversaries $\alpha < \frac{1}{2}$, the good workers will win the vote on average. To make this rigorous, we will need Lemma 1 and Cantelli's inequality. Cantelli (1928) tells us that for a random variable $X$ with mean $\mu$ and variance $\sigma^2$:

$$\mathbb{P}[\mu - X \geq |\lambda|] \leq \frac{1}{1 + \frac{\lambda^2}{\sigma^2}} \tag{1}$$

For a given gradient component, again let random variable $Z \in [0, M]$ denote the number of correct sign bits received by the parameter server. Let random variables $G$ and $B$ denote the number of good and bad workers (respectively) who (possibly inadvertently) sent the correct sign bit. Then, letting $p$ be the probability that a good worker computed the correct sign bit, $q := 1 - p$ and $\epsilon := p - \frac{1}{2}$ we can decompose $Z$ as follows:

$$Z = G + B$$
$$G \sim \text{binomial}[(1 - \alpha)M, p]$$
$$B \sim \text{binomial}[\alpha M, q]$$
$$\mathbb{E}[Z] = (1 - \alpha)Mp + \alpha Mq = \frac{M}{2} + (1 - 2\alpha)M\epsilon$$
$$\text{Var}[Z] = (1 - \alpha)Mpq + \alpha Mpq = M\left(\frac{1}{4} - \epsilon^2\right).$$

The vote only fails if $Z < \frac{M}{2}$ which happens with probability

$$\mathbb{P}\left[Z \leq \frac{M}{2}\right] = \mathbb{P}\left[\mathbb{E}[Z] - Z \geq \mathbb{E}[Z] - \frac{M}{2}\right]$$

$$\leq \frac{1}{1 + \frac{(\mathbb{E}[Z] - \frac{M}{2})^2}{\text{Var}[Z]}} \qquad \text{by Cantelli's inequality}$$

$$\leq \frac{1}{2}\sqrt{\frac{\text{Var}[Z]}{(\mathbb{E}[Z] - \frac{M}{2})^2}} \qquad \text{since } 1 + x^2 \geq 2x$$

$$= \frac{1}{2}\sqrt{\frac{M\left(\frac{1}{4} - \epsilon^2\right)}{(1 - 2\alpha)^2 M^2 \epsilon^2}}$$

$$= \frac{1}{2}\frac{\sqrt{\frac{1}{4\epsilon^2} - 1}}{(1 - 2\alpha)\sqrt{M}}$$

We now need to substitute in a bound on $\epsilon$. Assumption 4 and Lemma 1 tell us that

$$
\epsilon = \frac{1}{2} - q \geq \begin{cases} \frac{1}{2} - \frac{2}{9}\frac{1}{S^2} & \text{if } S > \frac{2}{\sqrt{3}}, \\ \frac{S}{2\sqrt{3}} & \text{otherwise.} \end{cases}
$$

From this we can derive that $\frac{1}{4\epsilon^2} - 1 < \frac{4}{S^2}$ as follows. First take the case $S \leq \frac{2}{\sqrt{3}}$. Then $\epsilon^2 \geq \frac{S^2}{12}$ and $\frac{1}{4\epsilon^2} - 1 \leq \frac{3}{S^2} - 1 < \frac{4}{S^2}$. Now take the case $S > \frac{2}{\sqrt{3}}$. Then $\epsilon \geq \frac{1}{2} - \frac{2}{9}\frac{1}{S^2}$ and we have $\frac{1}{4\epsilon^2} - 1 \leq \frac{1}{S^2}\frac{\frac{8}{9} - \frac{16}{81}\frac{1}{S^2}}{1 - \frac{8}{9}\frac{1}{S^2} + \frac{16}{81}\frac{1}{S^4}} < \frac{1}{S^2}\frac{\frac{8}{9}}{1 - \frac{8}{9}\frac{1}{S^2}} < \frac{4}{S^2}$ by the condition on $S$.

We have now completed the first part of the proof by showing the key statement that for the $i^{th}$ gradient component with signal to noise ratio $S_i := \frac{|g_i|}{\sigma_i}$, the failure probability of the majority vote is bounded by

$$
\mathbb{P}[\text{vote fails for } i^{th} \text{ coordinate}] = \mathbb{P}\left[Z_i \leq \frac{M}{2}\right] \leq \frac{1}{(1 - 2\alpha)\sqrt{M}S_i} \qquad (\star)
$$

The second stage of the proof will proceed by straightforwardly substituting this bound into the convergence analysis of SIGNSGD from Bernstein et al. (2018).

First let's bound the improvement of the objective during a single step of the algorithm for one instantiation of the noise. $\mathbb{I}[.]$ is the indicator function, $g_{k,i}$ denotes the $i^{th}$ component of the true gradient $g(x_k)$ and $\text{sign}(V_k)$ is the outcome of the vote at the $k^{th}$ iteration.

First take Assumption 2, plug in the step from Algorithm 1, and decompose the improvement to expose the error induced by stochasticity and adversarial workers:

$$
f_{k+1} - f_k \leq g_k^T(x_{k+1} - x_k) + \sum_{i=1}^{d}\frac{L_i}{2}(x_{k+1} - x_k)_i^2
$$

$$
= -\eta g_k^T \text{sign}(V_k) + \eta^2 \sum_{i=1}^{d}\frac{L_i}{2}
$$

$$
= -\eta\|g_k\|_1 + \frac{\eta^2}{2}\|\vec{L}\|_1 + 2\eta\sum_{i=1}^{d}|g_{k,i}|\,\mathbb{I}[\text{sign}(V_{k,i}) \neq \text{sign}(g_{k,i})]
$$

Next we find the expected improvement at time $k + 1$ conditioned on the previous iterate.

$$
\mathbb{E}[f_{k+1} - f_k|x_k] \leq -\eta\|g_k\|_1 + \frac{\eta^2}{2}\|\vec{L}\|_1 + 2\eta\sum_{i=1}^{d}|g_{k,i}|\,\mathbb{P}[\text{sign}(V_{k,i}) \neq \text{sign}(g_{k,i})]
$$

From $(\star)$, we have that the probability of the vote failing for the $i^{th}$ coordinate is bounded by

$$
\mathbb{P}[\text{sign}(V_{k,i}) \neq \text{sign}(g_{k,i})] \leq \frac{\sigma_{k,i}}{(1 - 2\alpha)\sqrt{M}|g_{k,i}|}
$$

where $\sigma_{k,i}$ refers to the variance of the $k^{th}$ stochastic gradient estimate, computed over a mini-batch of size $n$. Therefore, by Assumption 3, we have that $\sigma_{k,i} \leq \sigma_i/\sqrt{n}$.

We now substitute these results and our learning rate and mini-batch settings into the expected improvement:

$$
\mathbb{E}[f_{k+1} - f_k|x_k] \leq -\eta\|g_k\|_1 + \frac{2\eta}{\sqrt{n}}\frac{\|\vec{\sigma}\|_1}{(1 - 2\alpha)\sqrt{M}} + \frac{\eta^2}{2}\|\vec{L}\|_1
$$

$$
= -\sqrt{\frac{f_0 - f_*}{\|L\|_1 K}}\|g_k\|_1 + 2\sqrt{\frac{f_0 - f_*}{\|L\|_1 K^2}}\frac{\|\vec{\sigma}\|_1}{(1 - 2\alpha)\sqrt{M}} + \frac{f_0 - f_*}{2K}
$$

Now extend the expectation over randomness in the trajectory, and perform a telescoping sum over the iterations:

$$f_0 - f^* \geq f_0 - \mathbb{E}[f_K]$$

$$= \sum_{k=0}^{K-1} \mathbb{E}[f_k - f_{k+1}]$$

$$\geq \sum_{k=0}^{K-1} \mathbb{E}\left[ \sqrt{\frac{f_0 - f_*}{\|L\|_1 K}} \|g_k\|_1 - 2\sqrt{\frac{f_0 - f_*}{\|L\|_1 K^2}} \frac{\|\vec{\sigma}\|_1}{(1-2\alpha)\sqrt{M}} - \frac{f_0 - f_*}{2K} \right]$$

$$= \sqrt{\frac{K(f_0 - f_*)}{\|L\|_1}} \mathbb{E}\left[ \frac{1}{K} \sum_{k=0}^{K-1} \|g_k\|_1 \right] - 2\sqrt{\frac{f_0 - f_*}{\|L\|_1}} \frac{\|\vec{\sigma}\|_1}{(1-2\alpha)\sqrt{M}} - \frac{f_0 - f_*}{2}$$

We can rearrange this inequality to yield the rate:

$$\frac{1}{K} \sum_{k=0}^{K-1} \mathbb{E} \|g_k\|_1 \leq \frac{1}{\sqrt{K}} \left[ 2\frac{\|\vec{\sigma}\|_1}{(1-2\alpha)\sqrt{M}} + \frac{3}{2}\sqrt{\|L\|_1(f_0 - f^*)} \right]$$

$$\mathbb{E}\left[ \frac{1}{K} \sum_{k=0}^{K-1} \|g_k\|_1 \right] \leq \frac{1}{\sqrt{K}} \left[ \frac{3}{2}\sqrt{\|L\|_1}(f_0 - f_*) + 2\|\vec{\sigma}\|_1 \right]$$

Since we are growing our mini-batch size, it will take $N = O(K^2)$ gradient calls to reach step $K$. Substitute this in on the right hand side, square the result, use that $\frac{3}{2} < 2$, and we are done:

$$\left[ \frac{1}{K} \sum_{k=0}^{K-1} \mathbb{E} \|g_k\|_1 \right]^2 \leq \frac{4}{\sqrt{N}} \left[ \frac{\|\vec{\sigma}\|_1}{(1-2\alpha)\sqrt{M}} + \sqrt{\|L\|_1(f_0 - f^*)} \right]^2$$

$\square$