# OpenReview forum: "signSGD with Majority Vote is Communication Efficient and Fault Tolerant"
_ICLR.cc/2019/Conference_

### Official Review · AnonReviewer3 · 2018-11-02
**interesting distributed optimization algorithm based on signSGD**

**Rating:** 7
**Confidence:** 4

**Review:**

The paper proposes a distributed optimization method based on signSGD. Majority vote is used when aggregating the updates from different workers.
 The method itself is naturally communication efficient. Convergence analysis is provided under certain assumptions on the gradient. It also theoretically shows that it is robust up to half of the workers behave independently adversarially. Experiments are carried out on parameter server environment and are shown to be effective in speeding up training.

I find the paper to be solid and interesting. The idea of using signSGD for distributed optimization make it attractive as it is naturally communication efficient. The work provides theoretical convergence analysis under the small batch setting by further assuming the gradient is unimodal and symmetric, which is the main theoretical contribution. Another main theoretical contribution is showing it is Byzantine fault tolerant. The experiments are extensive, demonstrating running time speed-up comparison to normal SGD.

It is interesting to see a test set gap in the experiments. It remains to be further experimented to see if the method itself inherently suffer from generalization problems or it is a result of imperfect parameter tuning.

One thing that would be interesting to explore further is to see how asynchronous updates of signSGD affect the convergence both in theory and practice. For example, some workers might be lost during one iteration, how will this affect the overall convergence.
Also, it would be interesting to see the comparison of the proposed method with SGD + batch normalization, especially on their generalization performance. It might be interesting to explore what kind of regularization technique would be suitable for signed update kind of method.

Overall, I think the paper proposes a novel distributed optimization algorithm that has both theoretical and experimental contribution. The presentation of the paper is clear and easy to follow.

Suggestions: I feel the experiments part could still be improved as also mentioned in the paper to achieve competitive results. More experiments on different tasks and DNN architectures could be performed.

---

> ### Author Response · Authors · 2018-11-07
> **Response to AnonReviewer3**
>
> Dear AnonReviewer3,
>
> Thank you for your positive review. We really appreciate the remarks that our “experiments are extensive” and our paper is “solid and interesting”.
>
> >>>>>>>>>>>>>>>>>>>>>>>>>>>>
> >>>>>>> More experiments >>>>>>
> >>>>>>>>>>>>>>>>>>>>>>>>>>>>
>
> > “More experiments on different tasks and DNN architectures could be performed”
>
> Thanks for the suggestion, we have added experiments training the QRNN language model on the Wikitext-103 dataset. Please see the revisions above
>
> >>>>>>>>>>>>>>>>>>>>>>>>>>>>
> >>>>>>>> Further thoughts >>>>>>>
> >>>>>>>>>>>>>>>>>>>>>>>>>>>>
>
> > “some workers might be lost during one iteration”
> Intuitively, dropping workers will slow down convergence but not prevent it. You can see this immediately since a dropped worker is strictly better for convergence than an adversarial worker. This is one of the reasons we are excited about our Byzantine fault tolerance results.
>
> > what “regularization technique would be suitable for signed update kind of method”?
> We are particularly excited about this question for future work, thanks for suggesting it.

---

> ### Author Response · Authors · 2018-11-27
> **Revised the paper**
>
> Dear AnonReviewer3,
>
> We have updated your individualised response, and also summarised our revisions to the paper in a post above.
>
> Best wishes,
> AnonAuthors

---

### Official Review · AnonReviewer2 · 2018-11-03
**good work but can be improved**

**Rating:** 6
**Confidence:** 5

**Review:**

This paper continues the study of the signSGD algorithm due to (Balles & Hennig, Bernstein et al), where only the sign of a stochastic gradient is used for updating. There are two main results: (1) a slightly refined analysis of two results in Bernstein et al. The authors proved that signSGD continues to converge at the 1/sqrt(T) rate even with minibatch size 1 (instead of T as in Bernstein et al), if the gradient noise is symmetric and unimodal; (2) a similar convergence rate is obtained even when half of the worker machines flip the sign of their stochastic gradients. These results appear to be relatively straightforward extensions of those in Bernstein et al.

Clarity: The paper is mostly nicely written, with some occasionally imprecise claims.

Page 5, right before Remark 1: it is wrongly claimed that signSGD converges to a critical point of the objective. This cannot be inferred from Theorem 1. (If the authors disagree, please give the complete details on how the random sequence x_t converges to some critical point x^*. or perhaps you are using the word "convergence" differently from its usual meaning?)

Page 6, after Lemma 1. The authors claimed that "the bound is elegant since ... even at low SNR we still have ... <= 1/2." In my opinion, this is not elegant at all. This is just your symmetric assumption on the noise, nothing more...

Eq (1): are you assuming g_i > 0 here? this inequality is false as you need to discuss the two cases.

"Therefore signSGD cannot converge for these noise distributions, ..... point in the wrong direction." This is a claim based on intuitive arguments but not a proven fact. Please refrain from using definitive sentences like this.

Footnote 1: where is the discussion?


Originality: Compared to the existing work of Bernstein et al, the novelty of the current submission is moderate. The main results appear to be relatively straightforward refinements of those in Bernstein. The observation that majority voting is Byzantine fault tolerant is perhaps not very surprising but it is certainly nice to have a formal justification.

Quality: At times this submission feels like half-baked:
-- The theoretical results are about signSGD while the experiments are about sigNUM
-- The adversaries must send the negation of the sign? why can't they send an arbitrary bit vector?
-- From the authors' discussion " we will include this feature in our open source code release", "plan to run more extensive experiments in the immediate future and will update the paper...", and "should be possible to extend the result to the mini-batch setting by combining ..."

Significance: This paper is certainly a nice addition to our understanding of signSGD. However, the current obtained results are not very significant compared to the existing results: Theorem 1 is a minor refinement of the two results in Bernstein et al, while Theorem 2 at its current form is not very interesting, as it heavily restricts what an adversary worker machine can do. It would be more realistic if the adversaries can send random bits (still non-cooperated though).



##### added after author response #####
I appreciate the authors' efforts in trying to improve the draft by incorporating the reviewers' comments. While I do like the authors' continued study of signSGD, the submission has gone through some significant revision (more complete experiments + stronger adversary).

---

> ### Author Response · Authors · 2018-11-07
> **Response to AnonReviewer2**
>
> Dear AnonReviewer2,
>
> Thank you for your clear and thorough review. We appreciate your comment that the paper is a “nice addition to our understanding of signSGD”.
>
> We will first contest the criticism about the significance of the work. We will then respond to the other comments in detail.
>
> >>>>>>>>>>>>>>>>>>>>>>>>>>>>
> >>>> On matters of significance >>>
> >>>>>>>>>>>>>>>>>>>>>>>>>>>>
>
> > “it heavily restricts what an adversary worker machine can do”
> We have now formulated an entire class of adversaries that our algorithm is robust to. Please see our revisions above. This class contains machines that send random bits as a special case.
>
> > “Theorem 1 is a minor refinement”.
> Whilst "algebraically" the result is a minor refinement, conceptually it is a larger shift. It brings the signSGD work in line with modern machine learning practice. And we expect that it has ramifications on other active areas of ML research. For example:
>
> Reddi et al. (2018) showed how bimodal noise distributions can lead to divergence of Adam. This leaves a major outstanding question in the community: if Adam generally diverges, why does it work so well in practice? Theorem 1 shows how signSGD---a special limit of Adam---may be guaranteed to converge in natural settings such as Gaussian noise distributions. It suggests that we may be able to prove convergence of Adam for Gaussian noise distributions.
>
> >>>>>>>>>>>>>>>>>>>>>>>>>>>>
> >>>>>>> Minor comments >>>>>>>
> >>>>>>>>>>>>>>>>>>>>>>>>>>>>
>
> > “signSGD converges to a critical point of the objective”
> To clarify, we mean convergence in the sense that the gradient norm goes to zero as N increases, which is exactly what Theorem 1 tells us. Points with zero gradient norm are critical points. The mixed norm on the left hand side is unusual, but by inspection it is clear that the mixed norm shrinking to zero implies that the L2-norm shrinks to zero. We will clarify this in the paper.
>
> > “are you assuming g_i > 0 here”
> Thanks for mentioning this. We did not signpost it, but we assumed, without loss of generality, that g_i > 0. (The case that g_i < 0 follows by totally analogous reasoning.)
>
> > The claim “signSGD cannot converge for these noise distributions” is only “based on intuitive arguments”.
> Thank you for pointing this out, we decided to simplify the discussion by just giving a simple example.
>
> > ”The theoretical results are about signSGD while the experiments are about sigNUM”
> See [1, Appendix, Figure A.4] for experiments across a range of momentum values. [1] also discusses the theoretical relation between Signum and signSGD. In general we suggest practitioners use Signum instead of signSGD in practice since it is only fair to give our algorithm as many hyperparameters as momentum SGD.
>
> [1] signSGD, compressed optimisation for non-convex problems https://arxiv.org/abs/1802.04434.

---

> ### Author Response · Authors · 2018-11-27
> **Revised the paper**
>
> Dear AnonReviewer2,
>
> We have updated your individualised response, and also summarised our revisions to the paper in a post above.
>
> Best wishes,
> AnonAuthors

---

### Official Review · AnonReviewer1 · 2018-11-06
**A distributed implementation of signSGD with majority vote as aggregation. An interesting idea, that however is lacking comparisons with state of the art.**

**Rating:** 6
**Confidence:** 5

**Review:**

The authors present a distributed implementation of signSGD with majority vote as aggregation. The result is a  communication efficient and byzantine robust distributed training method. This is an interesting and relevant problem. There are two parts in this paper: first the authors prove a convergence guarantee for signSGD, and then they prove that under a weak adversary attack signSGD will be robust to a constant fraction of adversarial nodes. The authors conclude with some limited experiments.

Overall, the idea of combining low-communication methods with byzantine resilience is quite interesting. That is, by limiting the domain of the gradients one expects that the power of an adversary would be limited too. The application of the majority vote on the gradients is an intuitive technique that can resolve weak adversarial attacks. Overall, I found the premise quite interesting.

There are several issues that if fixed this could be a great paper, however I am not sure if there is enough time between rebuttals to achieve this for this round of submissions. I will summarize these key issues below.


1) Although the authors claim that this is a communication efficient technique, signSGD (on its communication merit) is not compared with any state of the art communication efficient training algorithm, for example:
- 1Bit SGD [1]
- QSD [2]
- TernGrad [3]
- Deep Gradient compression [4]
I think it is important to include at least one of those algorithms in a comparison. Due to the lack of comparisons with state of the art it is hard to argue on the relative performance of signSGD.

2) Although the authors claim byzantine resilience, this is against a very weak type of adversary, eg one that only sends back the opposite sign of the local stochastic gradient. An omniscient adversary can craft attacks that are significantly more sophisticated, for which a simple majority vote would not work. Please see the results in [b1].

3) The authors although reference some limited literature on byzantine ML, they do not compare with other byzantine tolerant ML methods. For example check [eg, b1-b4] below. Again, due to the lack of comparisons with state of the art it is hard to argue on the relative performance of signSGD.

Overall, although the presented ideas are promising, a substantial revision is needed before this paper is accepted for publication. I think it is extremely important that an extensive comparison is carried out with respect to both communication efficient algorithms, and/or byzantine tolerant algorithms, since signSGD aims to be competitive with both of these lines of work. This is a paper that has potential, but is currently limited by its lack of appropriate comparisons.



[1] https://www.microsoft.com/en-us/research/wp-content/uploads/2016/02/IS140694.pdf
[2] https://papers.nips.cc/paper/6768-qsgd-communication-efficient-sgd-via-gradient-quantization-and-encoding.pdf
[3] https://papers.nips.cc/paper/6749-terngrad-ternary-gradients-to-reduce-communication-in-distributed-deep-learning.pdf
[4] https://arxiv.org/pdf/1712.01887.pdf

[b1] https://arxiv.org/pdf/1802.07927.pdf
[b2] https://arxiv.org/pdf/1803.01498.pdf
[b3] https://dl.acm.org/citation.cfm?id=2933105
[b4] https://arxiv.org/pdf/1804.10140.pdf
[b5] https://arxiv.org/pdf/1802.10116.pdf

########################

I would like to commend the authors for making a significant effort in revising their manuscript. Specifically, I think adding the experiments for QSGD and Krum are an important addition. However, I still have a few major that in my opinion are significant:

- The experiments for QSGD are only carried for the 1-bit version of the algorithm. It has been well observed that this is by far the least well performing variant of QSGD. That is, 4 or 8 bit QSGD seems to be significantly more accurate for a given time budget. I think the goal of the experiments should not be to compare against other 1-bit algorithms (though to be precise, 1-bit QSGD is a ternary algorithm) , but against the fastest low-communication algorithm. As such, although the authors made an effort in adding more experiments, I am still not convinced that signSGD will be faster than 4 or 8 bit QSGD. I want to also acknowledge in this comment the fact that these experiments do take time, and are not easy to run, so I commend them again for this effort.

- My second comment relates to comparisons with state of the art algorithms in byzantine ML. The authors indeed did compare against Krum, however, as noted in my original review there are many works following Blanchard et al.

For example as I noted https://arxiv.org/pdf/1802.07927.pdf (the Bulyan algorithm) shows that there exist significantly stronger defense mechanisms for byzantine attacks. I think it would have been a much stronger comparison to compare with Bulyan.

Overall, I think the paper has good content, and the authors significantly revised their paper according to the reviews. However, several more experiments are needed for convincing a potential reader of the main claims of the paper, i.e., that signSGD is a state of the art communication efficient and byzantine tolerant algorithm.

I will increase my score from 5 to 6, and I will not oppose the paper being rejected or accepted. My personal opinion is that a resubmission for a future venue would yield a much stronger and more convincing paper assuming more extensive and thorough comparisons are added.

---

> ### Author Response · Authors · 2018-11-07
> **Response to AnonReviewer1**
>
> Dear AnonReviewer1,
>
> Thank you for your clear and precise review. We appreciate the comment that our work “could be a great paper” if we add some comparisons during the rebuttal. We want to contest your take on the weakness of our adversarial model, yet wholeheartedly agree with the need for adequate experimental comparisons to other techniques.
>
> >>>>>>>>>>>>>>>>>>>>>>>>>>>>
> >>>>>>> Comparison expts >>>>>>
> >>>>>>>>>>>>>>>>>>>>>>>>>>>>
>
> We have added comparisons to QSGD (compression) Multi-Krum (Byzantine fault tolerance). Please see the revisions in the post above.
>
> >>>>>>>>>>>>>>>>>>>>>>>>>>>>
> >>>>>>> Adversarial model >>>>>>
> >>>>>>>>>>>>>>>>>>>>>>>>>>>>
>
> > the adversary is “very weak” since it “only sends back the opposite sign of the local stochastic gradient”
> We have formulated an entire class of adversaries that our algorithm is robust to. Please see our revisions above.
>
> Thank you for pointing us to the paper [b1] saying that “convergence is not enough” since, for example, a powerful adversary can steer convergence to bad local minimisers. This is a great point. For this reason we do not recommend using our algorithm to protect against “omniscient” adversaries. But for “mere mortal” adversaries, our results are interesting. An example of a “mere mortal” adversary could be a broken machine that sends random bits or stale gradients.

---

> ### Author Response · Authors · 2018-11-27
> **Revised the paper**
>
> Dear AnonReviewer1,
>
> We have updated your individualised response, and also summarised our revisions to the paper in a post above.
>
> Best wishes,
> AnonAuthors

---

### Author Response · Authors · 2018-09-28
**Jupyter notebook**

Dear anonReviewers,

Here's a Jupyter notebook in case you'd like to play with the algorithm: https://colab.research.google.com/drive/1PlD2jXoXr2a8e57aIDINCw1-7RIttRTt

It can be run in the browser, or you can just download it and run on your machine.

Best wishes,
anonAuthors

---

### Author Response · Authors · 2018-11-07
**Summary of reviews**

Dear AnonReviewers,

Thank you for your thoughtful and thorough reviews. We will summarise the content of the reviews here.

First some high notes:

Rev3 says our “experiments are extensive” and our paper is “solid and interesting”. Rev2 says the paper is a “nice addition to our understanding of signSGD”. Rev1 says our work “could be a great paper” if we add sufficient comparisons during the rebuttal.

The reviewers’ main concerns:

1. Rev1 and Rev2 question the strength of the adversarial model;
2. Rev1 asks for comparison experiments for communication and/or Byzantine property;
3. Rev3 would like to see additional datasets and network architectures.

---

### Author Response · Authors · 2018-11-27
**Summary of revisions**

Dear AnonReviewers and AC,

We have updated the paper. The new version includes the following additions:

1. Added comparison to the Multi-Krum [1] Byzantine fault tolerant method (p9)
2. Added comparison to the QSGD [2] gradient compression method (p9)
3. Added natural language task benchmark (QRNN [3] model on the Wikitext-103 dataset) (p8)
4. Extended the robustness theorem to an entire class of adversaries that we term "blind multiplicative adversaries" (p7)

We are grateful to Rev1 and Rev3 for encouraging us to run the additional experiments, and for Rev2 for encouraging us to extend the robustness theorem.

We will now go into more detail:

1. Multi-Krum experiment. Multi-Krum is a Byzantine fault tolerant method that defines a security level f, and always removes f workers from the gradient aggregation (even when there are no adversaries present). Majority Vote in contrast always keeps all workers. We found that when the number of adversaries exceeds f, Multi-Krum deteriorates dramatically, whereas Majority Vote deteriorates more gracefully.

2. QSGD experiment. For a resnet-18 model on Cifar-10, we found that majority vote converges much faster than the "theory version" [2, p5] of the QSGD algorithm, but it converges at similar rate to the "experiment version" [2, p7] where the QSGD authors normalise by the max instead of the L2 norm. We found the max-norm version of QSGD had about 5x higher compression than the 32x compression of signSGD for this problem, but this gain represents a diminishing return since the cost of backpropagation has already started to dominate at that compression level.

To be explicit, for this network with SGD and NCCL, one epochs costs
=========> 6 sec computing + 12 sec communicating = 18 sec
For signSGD a very efficient implementation should reduce communication by 32x, therefore we expect one epoch to cost
=========> 6 sec computing + 12/32 sec communicating = 6.375 sec
For QSGD a very efficient implementation should reduce communication by (32x5)x, therefore one epoch should cost
=========> 6 sec computing + 12/(32x5) sec communicating = 6.075 sec
And we see the marginal gain of QSGD is small, whilst the algorithm is much more complicated.

3. Natural language experiment. We found that using signSGD with majority vote to train QRNN led to a 3x speedup per epoch over Adam with NCCL. That said, there was a deterioration in the converged solution. This meant that overall the performance after 2 hours of training was very similar.

4. Extended the robustness theorem. We show that Majority vote is robust to an entire class of adversaries that we call "blind multiplicative adversaries". This class includes adversaries that invert or randomise their gradient estimate as special cases.  We are particularly interested in randomised attacks as a model of network faults. This class of adversaries is more rigorous than the class of "non-cooperative" adversaries that we discussed previously.

[1] https://papers.nips.cc/paper/6617-machine-learning-with-adversaries-byzantine-tolerant-gradient-descent
[2] https://papers.nips.cc/paper/6768-qsgd-communication-efficient-sgd-via-gradient-quantization-and-encoding
[3] https://openreview.net/forum?id=H1zJ-v5xl

---

### Author Response · Authors · 2018-12-11
**Message to AC and AnonReviewer1**

Dear AC and AnonReviewer1,

The reviewers’ scores show a consensus to accept. Still, AnonReviewer1 raises important points that we want to address here.

1. QSGD precision. We agree, thanks for pointing it out. We are running experiments on 2 and 4bit QSGD and will add these to the paper.

2. Bulyan. We disagree. We believe this comparison is unnecessary for the following reasons:
———(A) our comparison with Krum is “good”—Krum successfully detects and eliminates the adversaries in our experiments. The only drawback of Krum is that is has a requirement for total num workers “n” to exceed num adversaries “f” by n > 2f + 2, therefore for n=7, Krum already breaks down at num adversaries f=3, whereas majority vote still works at f=3.
———(B) Bulyan, on the other hand, only tolerates up to 25% adversaries, requiring n > 4f + 3. For our case of 7 workers this means it only tolerates 1 adversary (f=1). Clearly Bulyan will perform worse than Krum on these experiments.
———(TL;DR) Krum already “aces” our experiments, except for the fact it has max security level f=2, therefore we didn’t see the need to compare to Bulyan which only serves to lower the max security level to f=1.
———(extra) There is another drawback of Krum and Bulyan, in that they throw away workers even when there are no adversaries—they have a “paranoid” regime. Majority vote does not do this. But this effect was not visible in our experiments (probably the batch size was too large to see it).

We therefore see no reason why the paper should not be accepted for this round of submission. In particular we think presenting the small batch theory (Theorem 1) would be an important and timely contribution to the understanding of adaptive gradient methods like Adam, which closely relate to signSGD. The paper may also spur further research into the combination of gradient compression and fault tolerance, which seem like a natural mix for large scale distributed learning.

Finally, we want to thank all the reviewers for their thorough, critical and constructive reviews.

---

> ### Author Response · Authors · 2018-12-21
> **We tested higher precision QSGD**
>
> Dear AC and AnonReviewer1,
>
> We have finished running 2, 4 and 8 bit QSGD. Per iteration, on our CIFAR-10 benchmark, we see:
>
> - max QSGD shows a tiny (insignificant) improvement at higher precision.
> - L2 QSGD shows larger improvement but is still roughly 2x slower than Majority Vote even at 8-bit precision.
>
> Therefore the claims in the paper still stand. We will add these results to the paper.

---

### Meta-Review · Area_Chair1 · 2018-12-12
**Good paper but requires revisions**

**Confidence:** 5
**Recommendation:** Accept (Poster)

**Metareview:**

The Reviewers noticed that the paper undergone many editions and raise concern about the content. They encourage improving experimental section further and strengthening the message of the paper.